# Coumarin Derivatives Inhibit ADP-Induced Platelet Activation and Aggregation

**DOI:** 10.3390/molecules27134054

**Published:** 2022-06-23

**Authors:** Ping-Hsun Lu, Tzu-Hsien Liao, Yau-Hung Chen, Yeng-Ling Hsu, Chan-Yen Kuo, Chuan-Chi Chan, Lu-Kai Wang, Ching-Yuh Chern, Fu-Ming Tsai

**Affiliations:** 1Department of Chinese Medicine, Taipei Tzu Chi Hospital, Buddhist Tzu Chi Medical Foundation, New Taipei City 231, Taiwan; pinghsunlu@gmail.com (P.-H.L.); a0958651263@gmail.com (T.-H.L.); 2School of Post-Baccalaureate Chinese Medicine, Tzu Chi University, Hualien 970, Taiwan; 3Department of Chemistry, Tamkang University, New Taipei City 251, Taiwan; yauhung@mail.tku.edu.tw; 4Department of Applied Chemistry, National Chia-Yi University, Chiayi City 600, Taiwan; s1100243@gmail.ncyu.edu.tw; 5Department of Research, Taipei Tzu Chi Hospital, Buddhist Tzu Chi Medical Foundation, New Taipei City 231, Taiwan; cykuo863135@gmail.com; 6Department of Laboratory Medicine, Taipei Tzu Chi Hospital, Buddhist Tzu Chi Medical Foundation, New Taipei City 231, Taiwan; kiki1205@tzuchi.com.tw; 7Department of Life Sciences, Ministry of Science and Technology, Taipei 106, Taiwan; keratin14kaikai@gmail.com

**Keywords:** coumarin, platelets, ADP, aggregation, 7-hydroxyflavone, flavonoids

## Abstract

Coumarin was first discovered in Tonka bean and then widely in other plants. Coumarin has an anticoagulant effect, and its derivative, warfarin, is a vitamin K analogue that inhibits the synthesis of clotting factors and is more widely used in the clinical treatment of endovascular embolism. At present, many artificial chemical synthesis methods can be used to modify the structure of coumarin to develop many effective drugs with low toxicity. In this study, we investigated the effects of six coumarin derivatives on the platelet aggregation induced by adenosine diphosphate (ADP). We found that the six coumarin derivatives inhibited the active form of GPIIb/IIIa on platelets and hence inhibit platelet aggregation. We found that 7-hydroxy-3-phenyl 4H-chromen-4-one (7-hydroxyflavone) had the most severe effect. In addition, we further analyzed the downstream signal transduction of the ADP receptor, including the release of calcium ions and the regulation of cAMP, which were inhibited by the six coumarin derivatives selected in this study. These results suggest that coumarin derivatives inhibit coagulation by inhibiting the synthesis of coagulation factors and they may also inhibit platelet aggregation.

## 1. Introduction

Coumarin, 1,2-benzopyrone, was originally isolated from *Melilotus officinalis* and is a natural substance in plants. Coumarin is widely found in plants, such as cinnamon, soybean sprouts, strawberries, and cherries [1]. Coumarin can be used as a spice and in medicine as a flavoring agent. The coumarin derivative warfarin is used to kill rodents, and this drug has anticoagulant functions [2,3,4]. Coumarin drugs are structurally similar to vitamin K. Coumarin-like drugs bind to vitamin K epoxide reductase complex 1 in the liver and block the conversion of inactive oxidative vitamin K into active reducing vitamin K. Active vitamin K is involved in the effects of coagulation factors II (reducing prothrombin production), VII, IX, and X. One side effect of warfarin is bleeding, which may be caused by its interaction with other drugs or food, resulting in enhanced anticoagulation effect [5]. To compensate for wound bleeding experienced with warfarin, researchers have studied coumarin derivatives to improve the ability of anticoagulants to treat blood clots and reduce the chance of bleeding.

Hemostasis is a process in which a series of reactions occur in the body to stop wound bleeding. The process of hemostasis is divided into three steps: vascular spasm, platelet plug formation, and clot formation. Coagulation is part of the hemostasis process. Coagulation (or clotting) is the process through which blood thickens into a gel-like consistency. This is the body’s way of stopping bleeding when needed. The coagulation cascade is divided into intrinsic and extrinsic pathways, both of which converge to a final common pathway to activate factor X, leading to fibrin formation. The intrinsic pathway consists of factors I, II, IX, X, XI, and XII. The extrinsic pathway consists of factors I, II, VII, and X. The common pathway consists of factors I, II, V, VIII, and X. Many factors circulate in the blood as zymogens and are activated into serine proteases. These serine proteases act as catalysts, cleaving the next zymogen into more serine proteases and finally activating fibrinogen. Researchers have explored the anticoagulant mechanism of coumarin derivatives primarily to inhibit the activation of coagulation factors [6,7,8,9,10].

Platelets are formed by the shedding of the cytoplasm of mature megakaryocytes in bone marrow, and each megakaryocyte produces 2000 to 7000 platelets [11]. When a wound occurs, the platelets gather at the damaged endothelium. Then, the platelets change shape, turn on receptors, and secrete chemical messengers. Finally, platelets bridge each other through receptors and aggregate [12,13,14]. The formation of platelet aggregates (primary hemostasis) is related to the activation of coagulation factors and the production, deposition and binding of fibrin (secondary hemostasis). These processes may overlap and result in the formation of either a white clot consisting primarily of platelet aggregates or a red clot consisting of platelet aggregates with fibrin clots and trapped red blood cells to stop bleeding [15]. Platelet aggregation is mediated by multiple pathways. Platelet activation by potent agonists such as thrombin or collagen causes the release of secondary agonists such as thromboxane A2 (TXA2) and the secretion of adenosine diphosphate (ADP) from platelet dense granules through a cyclooxygenase (COX)-dependent pathway [16]. The binding of TXA2 to the thromboxane receptor also leads to the secretion of ADP [17].

The above process results in the local accumulation of molecules such as TXA2 and ADP, which are important for further platelet recruitment and amplification of the aforementioned activation signals. Of these, ADP is the most important and well-studied physiological aggregator of platelets. The ADP-induced aggregation reaction is energy-consuming and dose-dependent, and it requires the participation of calcium ions and fibrinogen. ADP binds to P2Y1 and P2Y12 on the platelet membrane. Activation of the P2Y1 receptor leads to the hydrolysis of phosphoinositide and activates calcium ions to promote the formation of TXA2. Activation of the P2Y12 receptor inhibits platelet internal cyclic AMP (cAMP) activation [18,19,20]. Both of these mechanisms lead to platelet aggregation and activation of GPIIb/GPIIIa on the membrane [21,22].

Although coumarin is clinically used as an anticoagulant, the roles of coumarin and its derivatives in platelet aggregation have not been explored. We used ADP-induced human platelet aggregation to explore whether coumarin derivatives have antiplatelet aggregation functions. Our results indicated that coumarin derivatives have an antiplatelet aggregation effect. As such, in this study, we further found that coumarin derivatives block the signal transduction involved in the activation of P2Y1 and P2Y12 receptors by ADP. These results indicated that coumarin derivatives inhibit platelet activation in addition to having an inhibitory effect on the formation of coagulation factors.

## 2. Results

### 2.1. Coumarin Derivatives Inhibit ADP-Induced Platelet Aggregation

We synthesized six coumarin derivatives. Compounds 1, 2, and 4 are derivatives primarily synthesized from coumarin; compounds 3, 5, and 6 are derivatives synthesized primarily from isomers of coumarin (Figure 1). We first analyzed the effects of these compounds on ADP-induced human platelet aggregation. Under the action of a 10-ppm dose (ranging from 78 to 109.5 nM), various compounds inhibited the aggregation of platelets. They were all significant, and compound 5 had the best effect (Figure 2A). In addition to ADP, we also analyzed the effects of these compounds on other platelet agonists such as collagen-induced platelets. Similar results were observed, demonstrating that these compounds can significantly inhibit platelet aggregation induced by collagen (Appendix A). We further analyzed the dose effect of different compounds. All coumarin derivatives effectively inhibited human platelet aggregation at high doses (10–100 ppm). Compounds 3, 4, 5, and 6 also strongly inhibited the effect at a low dose (1 ppm) (Figure 2B). We performed subsequent experiments at a coumarin derivative concentration of 10 ppm.

### 2.2. Coumarin Derivatives Inhibit Expression of Glycoproteins on Platelet Membranes

GPIIb/IIIa is an integrin complex on the platelet membrane that participates in the binding of fibrinogen to von Willebrand factor and platelet activation. Therefore, we further analyzed whether coumarin derivatives also affected the expression of relevant glycoproteins on the platelet membrane while participating in platelet aggregation. As shown in Figure 3A,B, the addition of ADP slightly increased the expression of GPIIIa (CD61) on the platelet membrane and simultaneously induced the glycoprotein GPIIb/IIIa conformation change, resulting in the formation of an activated state that was recognized by the PAC-1 antibody. The addition of coumarin derivatives did not change the expression of ADP-induced CD61, but each coumarin derivative effectively inhibited the formation of the active form of GPIIb/IIIa complex (Figure 3B). These results suggested that the inhibition of platelet aggregation by coumarin derivatives is related to the inhibition of platelet membrane glycoprotein formation.

### 2.3. Coumarin Derivatives Inhibit Platelet Production of AA and TXA2

The glycoproteins on the membrane involved in the binding of platelets to fibrinogen trigger the aggregation of platelets with each other. Activated platelets secrete coagulation mediators, including ADP and TXA2, which leads to subsequent chain reactions. Therefore, we also explored whether coumarin derivatives affected the production of activated platelet TXB2 (the stable analogue of TXA2) and its upstream AA. Activation of platelets by fibrinogen and ADP substantially increased the production of AA and TXA2 in platelets (Figure 4A,B). The addition of various coumarin derivatives effectively inhibited the production of AA and TXA2 that had been induced by ADP. However, treatment of platelets with various coumarin derivatives had no effect on the production of AA or TXA2 (data not shown).

### 2.4. Coumarin Derivatives Inhibit Platelet ADP Receptor Signaling

The activation of fibrinogen receptors and the involvement of ADP in platelet aggregation both require the ADP receptor P2Y1 to participate in the influx of calcium ions and P2Y12 to participate in the negative regulation of cAMP. We first analyzed whether coumarin derivatives regulate platelet aggregation through the ADP receptor P2Y1 and P2Y12 signaling pathways. When platelets were treated with P2Y1 antagonist A2P5P and P2Y12 antagonist clopidogrel, coumarin derivatives could not further inhibit ADP-induced platelet aggregation (Appendix A). Additionally, we further analyzed whether the inhibition of ADP-induced platelet aggregation by coumarin derivatives is related to the blockade of ADP receptor signaling. The results in Figure 5A show that the ADP-stimulated platelets increased the calcium ion level in the platelets. The addition of each coumarin derivative remarkably inhibited the ADP-induced calcium influx in platelets. To observe the role of coumarin derivatives in platelet P2Y12 signaling, we first used the activator of adenylyl cyclase, forskolin, to activate the production of cAMP in platelets. The addition of ADP strongly inhibited the production of cAMP in platelets. When we added various coumarin derivatives, we did not observe the negative regulation of forskolin by ADP on cAMP production (Figure 5B), which indicated that coumarin derivatives blocked the signal transduction of the ADP receptor P2Y12. The results from Figure 5B show that coumarin derivatives appear to increase cAMP production in platelets. However, when platelets were only treated with coumarin derivatives, the production of cAMP did not change significantly compared with the control group not treated with forskolin or ADP, suggesting that coumarin derivatives have no effect on resting platelets (data not shown).

## 3. Discussion

Coumarin drugs are synthesized from coumarin, one of the important derivatives of which is warfarin. The anticoagulant effect of coumarin is based on the activity of its derivatives (warfarin or dicumarol). The anticoagulant effect of coumarin may be achieved by participating in the antagonism of vitamin K metabolism [23,24]. Whether the compound we used in our study that we derived from coumarin or from coumarin isomer has the same antagonistic function on vitamin K metabolism as warfarin is still unknown. Although whether these compounds have an anti-vitamin K metabolism effect is still unknown, we found that several compounds could prevent phenylhydrazine-induced hemolytic anemia in in vivo experiments with zebrafish embryos, suggesting that they can inhibit the formation of thrombus (data not shown).

The main cause of ischemic stroke is cerebral vascular obstruction. Coumarin derivatives such as warfarin are primarily used in the clinic for ischemic stroke patients with atrial fibrillation or artificial heart valves to inhibit the synthesis of related coagulation factors and achieve anticoagulant effects [25,26]. The main function of platelets is to stop bleeding. When a blood vessel ruptures, platelets clump to stop the bleeding. However, even minor endothelium damage leads to the gradual accumulation of platelets after vascular sclerosis, which gradually narrows the blood vessel and eventually blocks the whole blood vessel. Therefore, inhibiting the activity of platelets can also reduce the chance of a blood vessel blockade. Aspirin is the antiplatelet agent that has been used for the longest time. Antiplatelet drugs, such as aspirin, or anticoagulant drugs, such as warfarin, can reduce the chance of cerebral vascular obstruction and prevent another stroke. Six coumarin derivatives inhibited the aggregation of platelets in our study; although the inhibitory effects were weak, their effects were significant different. These results suggested that many of the clinical side effects of coumarin derivatives may lie in their effect on more than just the formation of coagulation factors. However, as mentioned above, whether a single coumarin derivative has multiple simultaneous targets in anticoagulation still needs to be further verified.

In this study, we used only ADP as the inducer to analyze the inhibition of platelet aggregation by coumarin derivatives. Comparing coumarin derivatives with commonly used P2Y1 antagonists, such as MRS2279, and P2Y12 antagonists, such as clopidogrel, the IC50s of MRS2279 and clopidogrel are 100–300 nM and 1.9 μM, respectively [27,28]. Although the ability of coumarin derivatives to inhibit ADP-induced platelet aggregation estimated from dose use is similar to that of the aforementioned antagonists, coumarin derivatives less specifically inhibit the signaling of ADP receptors: both P2Y1 and P212 receptors are affected by coumarin derivatives. In addition to ADP, epinephrine, thromboxane A2, and thrombin can be used as agonists to promote platelet aggregation. All of the above agonists participate in platelet aggregation through the production of ADP [16,17]. Therefore, we deduced that, even if the addition of the above agonists induces platelet aggregation, coumarin derivatives may have inhibitory effects similar to collagen as an agonist (Appendix A), but further experiments are needed to confirm this hypothesis. Furthermore, although coumarin derivatives inhibit platelet aggregation by inhibiting the signal transduction of P2Y1 and P2Y12 receptors when we analyzed ADP receptor antagonists (Appendix A), other unknown effects of coumarin derivatives beyond those observables in response to ADP cannot be excluded.

The plasma levels of warfarin and 7-hydroxywarfarin in patients receiving warfarin were 11.4 and 3.7 μM, respectively [29]; these results suggested that the dose of coumarin we used in this study is within a reasonable range. Our results showed that compound 5 had the strongest antiplatelet aggregation activity among the several coumarin derivatives. Flavonoids have an antiplatelet aggregation effect [30,31,32,33], and 7-hydroxyflavone inhibits COX-1, which inhibits TXA2 production in platelets and platelet aggregation [34]. We obtained similar results in our experiments, and we further explored whether these coumarin derivatives block the P2Y1 and P2Y12 signaling pathways. No literature exists on the role of vitamin K in platelet aggregation. These results suggested that the inhibition of platelet aggregation by coumarin derivatives is related to these derivatives being structurally similar to flavonoids and not vitamin K.

In addition to the ADP receptor involved in platelet aggregation, other functions such as P2Y12 are involved in the release of platelet granules or apoptosis via the PI3K/AKT pathway [35]. The involvement of AKT activation in platelet apoptosis was verified by analyzing platelet populations in a flow cytometer [36]. However, when analyzing the expression of glycoproteins on platelet membranes, coumarin derivatives did not substantially alter the platelet population, suggesting that coumarin derivatives do not affect the PI3K/AKT pathway in platelets. These results showed that coumarin derivatives affect platelet aggregation mainly through the inhibition of platelet activation rather than by affecting its apoptosis. Among the coumarin derivatives we studied, no literature exists on whether compounds 1, 2, 3, 4, and 6 are natural products or potential functions of biological metabolism for these five compounds. Compound 5, also known as 7-hydroxyflavone, is mostly one of the secondary metabolites of phenylalanine, which is metabolized by plant organisms, and is one of the flavonoids, which may be converted into other products by microorganisms [37]. It is known that 7-Hydroxyflavone has antiviral [38], anti-inflammatory [39], and antidrug-resistant cancer cell growth [40] properties. These activities of 7-hydroxyflavone may be related to the common activity of flavonoids [41]. Additionally, 7-Hydroxyflavone is a potential inhibitor of CYP1A1 per structure analysis, and this inhibition of CYP1A1 activity may also be involved in many physiological functions.

In summary, we found that several chemosynthetic coumarin derivatives inhibited ADP-regulated human platelet aggregation. The downstream calcium ion production of P2Y1 and the negative regulation of cAMP participation of P2Y12 may be inhibited by coumarin derivatives, which indicates that coumarin derivatives comprehensively inhibit signal transduction within platelets or blocks bridge communication between platelets. The antiplatelet aggregation activity of coumarin derivatives may be related to their structures, which contain flavonoids.

## 4. Materials and Methods

### 4.1. Coumarin Derivatives

Our process of synthesizing coumarin derivatives is shown in Appendix A. We confirmed all of the synthesized compounds using NMR spectroscopy. We recorded proton (300 MHz) and carbon (75 MHz) NMR spectra on a Varian Mercury-300 NMR spectrometer (Agilent, Santa Clara, CA, USA). Chemical shifts are reported on the δ scale as parts per million (ppm) downfield from tetramethylsilane (TMS) as an internal reference. We used all commercial reagents as obtained. The spectra of various compounds we used in this study are as follows:

7-[(2-methylbut-3-yn-2-yl)oxy]-2H-chromen-2-one (compound 1)

^1^H-NMR (CDCl3, 300 MHz): 7.64 (d, J = 9.6 Hz, 1H), 7.36 (d, J = 8.8 Hz, 1H), 7.04 (dd, J = 2.0, 8.8 Hz, 1H), 6.27 (d, J = 9.6 Hz, 1H), 2.66 (s, 1H), 1.71 (s, 6H)

^13^C-NMR (CDCl3, 75 MHz): 161.2, 159.1, 155.0, 143.3, 128.2, 117.0, 113.7, 113.5, 107.1, 84.6, 75.2, 72.8, 29.6.

8,8-dimethyl-2H,8H-pyrano [2,3-f]chromen-2-one (compound 2)

^1^H-NMR (CDCl3, 300 MHz): 7.61 (d, J = 9.6 Hz, 1H), 7.21 (d, J = 8.8 Hz, 1H), 6.87 (d, J = 10.0 Hz, 1H), 6.72 (d, J = 8.8 Hz, 1H), 6.23 (d, J = 9.6 Hz, 1H), 5.72 (d, J = 10.0 Hz, 1H), 1.47 (s, 6H)

^13^C-NMR (CDCl3, 75 MHz): 161.4, 156.1, 150.0, 144.3, 130.7, 127.8, 114.7, 113.5, 112.6, 112.4, 109.3, 77.6, 28.2.

8,8-dimethyl-2H,8H-pyrano [3,2-g]chromen-2-one (compound 3)

^1^H-NMR (CDCl3, 300 MHz): 7.59 (d, J = 9.5 Hz, 1H), 7.27 (s, 1H), 6.72 (s, 1H), 6.34 (d, J = 10.0 Hz, 1H), 6.22 (d, J = 9.5 Hz, 1H), 5.69 (d, J = 10.0 Hz, 1H), 1.47 (s, 6H)

^13^C-NMR (CDCl3, 75 MHz): 161.3, 156.9, 155.4, 143.4, 131.2, 124.8, 120.8, 118.6, 113.0, 112.7, 104.4, 109.3, 77.8, 28.2.

7-hydroxy-3-phenyl-4H-chromen-4-one (compound 4)

^1^H-NMR (D6-DMSO, 300 MHz): 10.63 (s,1H, -OH), 8.15 (s, 1H), 7.68 (dd, J = 8.7, 1.8 Hz, 2H), 7.60 (d, J = 8.7 Hz, 1H), 7.46–7.37 (m, 3H), 6.82 (dd, J = 8.7, 1.8 Hz, 1H), 6.75 (d, J = 1.8 Hz, 1H).

^13^C-NMR (D6-DMSO, 75 MHz): 174.4, 162.7, 157.5, 153.8, 132.0, 129.1, 128.2, 127.7, 127.3, 123.5, 116.6, 115.3, 102.2.

2-(2,2-dimethyl-2H-chromen-6-yl)-7-hydroxychroman-4-one (compound 5)

^1^H-NMR (CDCl3, 300 MHz): 7.85 (d, J = 8.8 Hz, 1H), 7.18 (dd, J = 8.4, 2.4 Hz, 1H), 7.08 (d, J = 2.4 Hz, 1H), 6.81 (d, J = 8.4 Hz, 1H), 6.54 (dd, J = 8.8, 2.4 Hz, 1H), 6.44 (d, J = 2.4 Hz, 1H), 6.33 (d, J = 9.8 Hz, 1H), 6.02 (s,1H, -OH), 5.65 (d, J = 9.8 Hz, 1H), 5.35 (dd, J = 13.2, 2.8 Hz, 1H), 3.04 (dd, J = 13.2, 16.8 Hz, 1H), 3.04 (dd, J = 16.8, 2.8 Hz, 1H), 1.44 (s, 6H).

^13^C-NMR (CDCl3, 75 MHz): 191.9, 163.9, 163.7, 153.4, 131.4, 130.7, 129.4, 127.2, 124.4, 121.9, 121.4, 116.5, 116.1, 114.5, 110.8, 103.4, 79.6, 43.9, 28.1.

HRMS calculated for C20H18O4: 322.1205; found: 322.1211.

7-hydroxy-2′,2′-dimethyl-2′H,4H-[3,6′-bichromen]-4-one 1 (compound 6)

^1^H-NMR (CD3OD, 300 MHz): 8.17 (s,1H), 8.06 (d, J = 8.8 Hz, 1H), 7.27 (dd, J = 8.8, 2.0 Hz, 1H), 7.21 (d, J = 2.0 Hz, 1H), 6.94 (dd, J = 8.8, 2.0 Hz, 1H), 6.86 (d, J = 2.0 Hz, 1H), 6.78 (d, J = 8.8 Hz, 1H), 6.41 (d, J = 9.8 Hz, 1H), 5.73 (d, J = 9.8 Hz, 1H), 4.62 (s,1H, -OH), 1.43 (s, 6H).

^13^C-NMR (CDCl3, 75 MHz): 178.1, 164.8, 159.9, 155.0, 154.4, 132.4, 131.0, 128.6, 128.3, 125.8, 125.7, 123.3, 122.7, 118.3, 117.3, 116.6, 103.3, 77.6, 28.0.

### 4.2. Preparation of Platelet-Rich Plasma (PRP), Platelet-Poor Plasma (PPP), and Purified Platelets

Human blood was obtained from healthy adults via vacuum venipuncture. All participants in the study understood the purpose of the study and consented to blood sample collection. The Taipei Tzu Chi Hospital Institutional Review Board approved this study. We obtained PRP from the supernatant obtained from centrifugation at 180× *g* for 20 min at room temperature. We centrifuged PRP at 1500× *g* for 15 min to obtain PPP from supernatants, and we purified platelets by resuspension in modified calcium-free Tyrode buffer.

### 4.3. Measurement of Platelet Aggregation

We measured platelet aggregation using a transmittance aggregometer platform (PAP-8E, Platelet Aggregation Profiler, Bio/Data Corporation, Horsham, PA, USA). We obtained 400 μL of PPP from an individual, which we used as the clear reference value of individual plasma for calibration regression. We treated each 400 μL of PRP with 1 μL of coumarin derivatives (the 10 ppm doses of the final concentrations of each derivative were 109.5, 109.5, 78 3, 109.5, 105, and 77.6 nM for compounds 1–6, respectively) in the experimental group, which we placed on the transmittance aggregator platform at 37 °C for 30 min. We moved samples to the monitoring area of the transmittance aggregometer platform to start transmittance monitoring. We added ADP (10 μM, Sigma-Aldrich, St. Louis, MO, USA) and observed plasma transmittance for 6 min to determine the platelet aggregation status.

### 4.4. Measurement of Platelet Surface Glycoproteins

We diluted the obtained blood 1:2 with PBS. We treated diluted blood (50 μL) with 10 ppm of coumarin derivatives for 30 min at 37 °C, followed by fluorescent antibodies recognizing CD61 (clone: VI-PL2, PE-conjugated, BD Pharmingen, San Jose, CA, USA), activated GP IIb-IIIa complex (clone: PAC-1, FITC-conjugated, BD Pharmingen), and 20 μM ADP for 30 min. We immediately fixed the diluted blood samples by adding 400 μL of 1% paraformaldehyde/PBS. We analyzed the expression of platelet surface antigen using flow cytometry (FACScan, Becton Dickinson, Franklin Lakes, NJ, USA).

### 4.5. Measurement of Arachidonic-Acid (AA) Release from Platelets

We treated human PRP with 10 ppm of coumarin derivatives for 1 h. We further centrifuged PRP-treated with coumarin derivatives to obtain purified platelets. We treated purified platelets with 10 μM ADP and 3 μM fibrinogen for 3 min at 37 °C. We detected arachidonic acid in the supernatant using an EIA kit (Cusabio Biotech, Wuhan, China).

### 4.6. Measurement of Thromboxane A2 (TXA2) in Platelets

We treated human PRP with 10 ppm of coumarin derivatives for 1 h. After we obtained the purified platelets via centrifugation, we incubated them with 3 μM fibrinogen and 10 μM ADP at 37 °C for 3 min. We terminated the reaction by placing the tube in an environment containing liquid nitrogen to quickly freeze the sample. We thawed the samples at room temperature, and the supernatant was obtained via centrifugation at 3000× *g* at 4 °C for 10 min. The stable metabolite TXB2 of TXA2 contained in the supernatant was detected using an EIA kit (Cayman Chemical, Ann Arbor, MI, USA).

### 4.7. Measurement of Cytosolic Free Calcium in Platelets

We treated human PRP with 10 ppm of coumarin derivatives for 1 h, followed by 3 μM Fura-2AM (Molecular Probes, Invitrogen, Carlsbad, CA, USA) at 37 °C for 45 min. We obtained purified platelets via centrifugation, which we suspended in modified calcium-free Tyrode buffer. We stimulated purified platelets with 10 μM ADP, and we immediately measured the amount of fluorescence using a multiplate reader (Infinite F200, Tecon, Durham, NC, USA) at 37 °C at excitations of 340 and 380 nm and emission of 510 nm.

### 4.8. Measurement of cAMP Content in Platelets

We treated human PRP with 10 ppm of coumarin derivatives for 1 h. We suspended purified platelets obtained via centrifugation in modified calcium-free Tyrode buffer, which we then incubated with 10 μM forskolin and 10 μM ADP at 37 °C for 5 min. After washing twice with PBS, we lysed the cells with 0.1 N HCl and collected the supernatant via centrifugation. We detected the cAMP content in the supernatant using an EIA kit (Cayman Chemical).

### 4.9. Statistical Analysis

We obtained the data for each experiment from at least three samples. We performed experiments in triplicate, and the results are expressed as mean ± SD. We performed statistical analyses using one-way ANOVA with Dunnett’s post-test. We considered a between-group *p* value of <0.5 to indicate statistical significance.

## 5. Conclusions

Our results indicated that the six tested coumarin derivatives inhibited ADP-induced platelet aggregation. Coumarin derivatives also affected the expression of glycoproteins GPIIb/IIIa on activated platelet membranes involved in binding to other molecules and the release of TXA2. We found that coumarin derivatives inhibited the activation of calcium ions and the regulation of cAMP downstream of the signaling pathways of ADP receptors P2Y1 and P2Y12. Therefore, the results suggested that coumarin derivatives inhibit platelet activation by blocking the ADP signaling pathway.

## Figures and Tables

**Figure 1 molecules-27-04054-f001:**
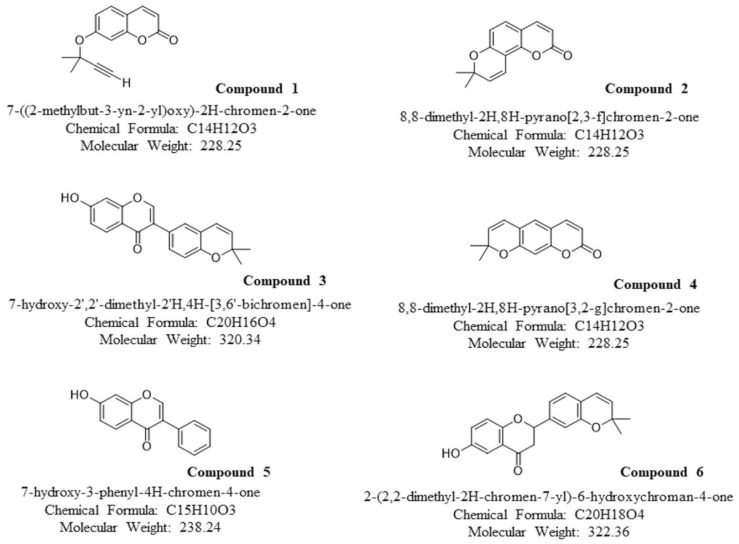
Chemical structure of the different assayed coumarin derivatives.

**Figure 2 molecules-27-04054-f002:**
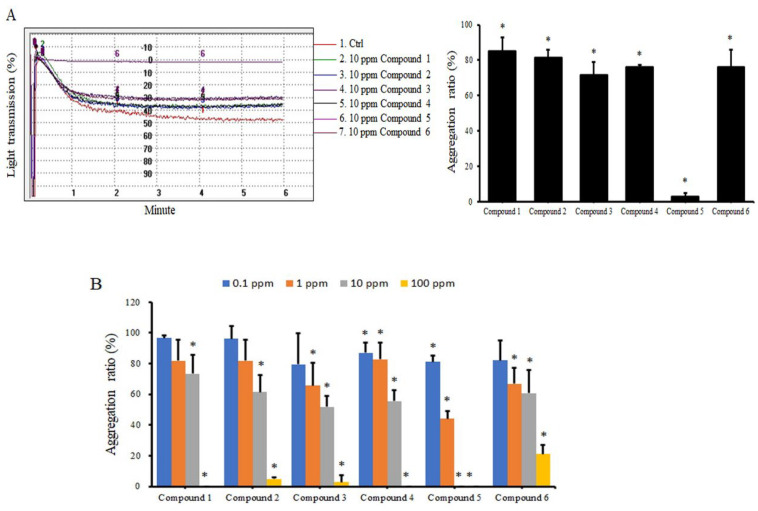
Effects of coumarin derivatives on ADP−induced platelet aggregation. Human PRP was treated with DMSO (control) or 10 ppm of coumarin derivates (**A**) or the indicated concentrations of coumarin derivatives (**B**) for 30 min, followed by addition of 10 μM ADP, and aggregation was analyzed using an aggregometer for 6 min. The data are presented as the aggregation ratio relative to control group treated with ADP only (*n* = 3). * *p* < 0.05 compared with the control group treated with ADP only.

**Figure 3 molecules-27-04054-f003:**
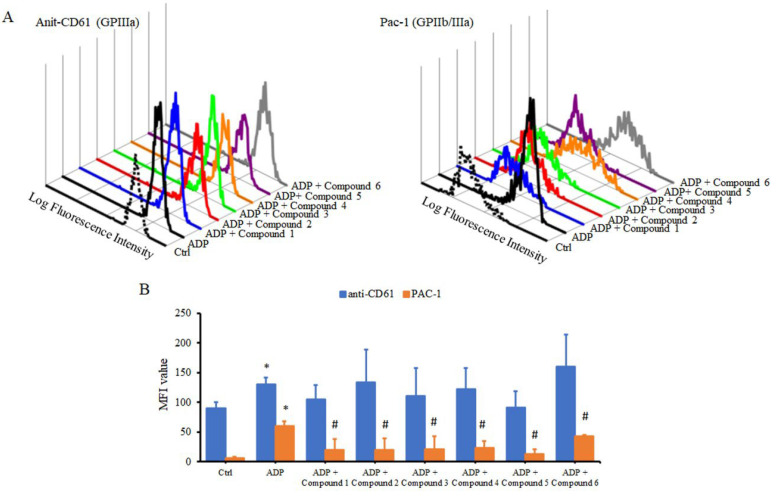
Effects of coumarin derivatives on the expression of glycoproteins on human platelet membranes. Human blood was treated with 10 ppm of coumarin derivatives for 30 min, followed by the addition of 20 μM ADP for 30 min. The expression of GPIIIa and active form of GPIIb/IIIa on the membrane was analyzed using flow cytometry. The flow data for GPIIIa and GPIIb/IIIa on platelet membranes from one of three independent experiments (*n* = 3) (**A**). Experimental results are summarized as MFI values of platelet membrane (**B**). * *p* < 0.05 compared with the control group. # *p* < 0.05 compared with whole blood stimulated with ADP alone.

**Figure 4 molecules-27-04054-f004:**
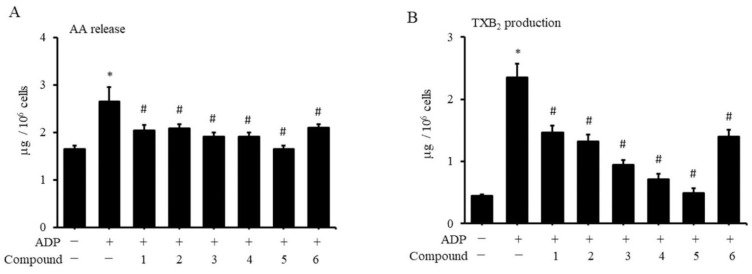
Effects of coumarin derivatives on the release of AA and TXA2 from human platelets. Human PRP was treated with 10 ppm of coumarin derivatives for 1 h. Purified plates were isolated, and 10 μM ADP and 3 μM fibrinogen were added for 30 min. Contents of AA (**A**) and TXA2 (**B**) were analyzed using an enzyme immunoassay (*n* = 3). * *p* < 0.05 compared with the control group. # *p* value < 0.05 for PRP treated with indicated compound followed by ADP stimulation compared with PRP stimulated with ADP alone.

**Figure 5 molecules-27-04054-f005:**
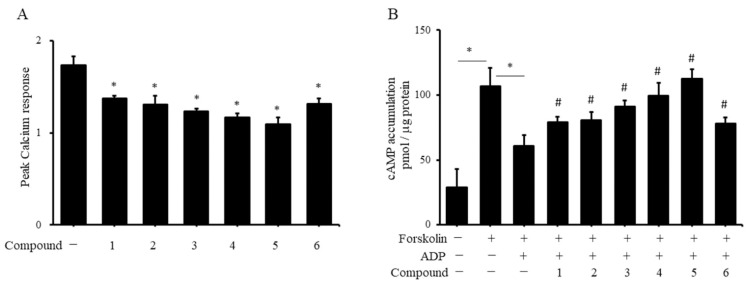
Effects of coumarin derivatives on calcium and cAMP production in human platelets. Human PRP was treated with 10 ppm of coumarin derivatives for 1 h, and Fura-2AM reagent was added. Purified plate isolates were treated with 10 μM ADP, and calcium concentration was monitored in an EIA reader. Data are presented as the fold change relative to control group not treated with ADP (**A**). * *p* < 0.05 compared with control group. Human PRP was treated with 10 ppm of coumarin derivatives for 1 h, and isolated platelets were treated with 10 μM forskolin and 10 μM ADP for 5 min. Platelet lysates were prepared, and cAMP levels were measured using an enzyme immunoassay (*n* = 3). * *p* < 0.05 between two groups. # *p* < 0.05 for PRP treatment of indicated compound followed by forskolin and ADP stimulation compared with PRP stimulated with forskolin and ADP alone (**B**).

## Data Availability

Not applicable. All data used to support the results of this study are included in the article.

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
