# Peer review of "Coumarin Derivatives Inhibit ADP-Induced Platelet Activation and Aggregation"

_molecules, 2022, doi:10.3390/molecules27134054_

Round 1
Reviewer 1 Report
In this manuscript, Lu et al describe the action of coumarin derivatives beyond coagulation, inhibiting ADP-mediated platelet activation.
The subject is interesting in the field, especially nowadays that there is general concern on the adverse events of anti-coagulant or anti-platelet agents (i.e. bleeding).
There are some issues of concern and minor things to be corrected, I go by section:
Abstract: line 27: GPIIB/IIIA formation on platelet membrane: should be corrected to "inhibit the active form of GPIIB/IIIA on platelets, and hence, inhibit platelet aggregation", as one follows the other, becaose the lack of the active form will result in reduced fibrinogen platelet-platelet bridges.
Line 28: an inhibitor has the best effect... change to "the most acute or most severe effect..."
Line 32: coumarin may be involved in platelet aggregation... change to may inhibit...
Introduction:
Line 44: antihemophilic agent?? please rename, as this leads to misinterpretation.
Results: Line 102: The effect of compound 5 was the most significant... change to the lowest, as they are all significant.
MAJOR POINT
Figure 3, and lines 115-116: It is impossible that platelets are CD61 negative (Control). They should be all positive. The difference in expression levels upon activation, might be due to degranulation and translocation of receptors found in granules, into the platelet membrane. But all platelets should be CD61 positive prior stimulation.
After stimulation, the majority should be positive, and some negativity may be measured if platelets have vesiculated, but overall, platelets should also be all positive for CD61, and maybe shift a little bit upon stimulation, but not the two peaks shown. How are the platelets gated?
The binding of PAC1, tells that the integrin GPIIB/IIIA is in the active form (it is not a PAC-1 complex, it is an antibody binding only to the active form)
Ideally, controls should be CD61 positive and PAC-1 negative, while platelets upon ADP should be both CD61 and PAC1 positive, and the effect of the coumarin derivatives should then studied between these two extremes, if any effect at all is exerted.
Author Response
Response Reviewer 1:
Reviewer 1
In this manuscript, Lu et al describe the action of coumarin derivatives beyond coagulation, inhibiting ADP-mediated platelet activation.
The subject is interesting in the field, especially nowadays that there is general concern on the adverse events of anti-coagulant or anti-platelet agents (i.e. bleeding).
There are some issues of concern and minor things to be corrected, I go by section:
Abstract: line 27: GPIIB/IIIA formation on platelet membrane: should be corrected to "inhibit the active form of GPIIB/IIIA on platelets, and hence, inhibit platelet aggregation", as one follows the other, becaose the lack of the active form will result in reduced fibrinogen platelet-platelet bridges.
Reply: Thanks for the reviewer's suggestion. We have made changes to the text according to the reviewer’s comment (line 27-28).
Line 28: an inhibitor has the best effect... change to "the most acute or most severe effect..." Reply: Thanks for the reviewer's suggestion. We have made changes to the text according to the reviewer’s comment (line 28-29).
Line 32: coumarin may be involved in platelet aggregation... change to may inhibit...
Reply: Thanks for the reviewer's suggestion. We have made changes to the text according to the reviewer’s comment (line 33).
Introduction:
Line 44: antihemophilic agent?? please rename, as this leads to misinterpretation.
Reply: Thanks for the reviewer's suggestion. This content has been removed due to substantial revision of the manuscript content.
Results: Line 102: The effect of compound 5 was the most significant... change to the lowest, as they are all significant.
Reply: Thanks for the reviewer's suggestion. We have made changes to the text according to the reviewer’s comment (line 104-105).
MAJOR POINT
Figure 3, and lines 115-116: It is impossible that platelets are CD61 negative (Control). They should be all positive. The difference in expression levels upon activation, might be due to degranulation and translocation of receptors found in granules, into the platelet membrane. But all platelets should be CD61 positive prior stimulation.
After stimulation, the majority should be positive, and some negativity may be measured if platelets have vesiculated, but overall, platelets should also be all positive for CD61, and maybe shift a little bit upon stimulation, but not the two peaks shown. How are the platelets gated?
The binding of PAC1, tells that the integrin GPIIB/IIIA is in the active form (it is not a PAC-1 complex, it is an antibody binding only to the active form)
Ideally, controls should be CD61 positive and PAC-1 negative, while platelets upon ADP should be both CD61 and PAC1 positive, and the effect of the coumarin derivatives should then studied between these two extremes, if any effect at all is exerted.
Reply: Thank you reviewer for your professional advice. The method used in the 1st manuscript was to perform preliminary screening based on cell size populations and directly analyze anti-CD61 and PAC-1. We re-performed this experiment with minor modifications, selecting on the basis of cell size, and the distribution of CD61 expression still had two populations (fewer populations on the left). Based on the distribution of isotype antibody staining, we selected the expression groups of CD61 for subsequent analysis, as shown in the figure below. The experimental results showed that the addition of ADP still induced the expression of CD61 on the platelet membrane, while the coumarin derivatives inhibited the activated state of GPIIb/IIIa on the platelet membrane. Figure 3 was the revised version, with the relevant text in the Results section (line 119-130).

Reviewer 2 Report
The authors report that chemosynthetic coumarin derivatives, well-known vitamin-K antagonists, inhibited ADP-regulated human platelet aggregation. Furthermore, ADP P2Y1 receptor-dependent calcium release and ADP P2Y12 receptor-dependent down-regulation of cAMP was inhibited by the studied coumarin derivatives. The authors suggest that the antiplatelet aggregation activity of coumarin derivatives may be related to their structures containing flavonoids.
The experiments appear to be well done, and the data are of some interest. However, there are some major questions, which need to be addressed by the authors, also by additional data/experiments.
11) It is not clear what the overall aim of this study is. Do the authors aim to have novel compounds which inhibit both vitamin K-effects and platelets? Do the newly synthesized coumarins still have vitamin K-antagonist properties?
22) Concentrations of compounds used. While the authors report the concentrations of ADP and fibrinogen in normal biochemical terms (i.e. µM), concentration of the coumarin derivatives is reported in ppm. For direct comparison, the concentration of coumarins should be given also in µM.
33) Medical/pharmacological relevance. Multiple chemical compounds unspecifically inhibit platelet activation by ADP and other agonists, especially at concentrations higher than 5-10 µM. The first question is whether pharmacologically used coumarins reach plasma concentrations in the rage of the coumarin concentration used here ? Then, dose response curve would be helpful, ideally in comparison with established ADP receptor antagonists.
44) Specificity : Do the coumarin derivatives affect only platelet ADP receptors? The authors should also study other platelet agonists (thromboxane A2, thrombin, collagen etc.)
55) P2Y12 receptor responses include much more than down-regulation of cAMP, i.e. PI3K responses.
Author Response
Response to Reviewer 2:
Reviewer 2:
The authors report that chemosynthetic coumarin derivatives, well-known vitamin-K antagonists, inhibited ADP-regulated human platelet aggregation. Furthermore, ADP P2Y1 receptor-dependent calcium release and ADP P2Y12 receptor-dependent down-regulation of cAMP was inhibited by the studied coumarin derivatives. The authors suggest that the antiplatelet aggregation activity of coumarin derivatives may be related to their structures containing flavonoids.
The experiments appear to be well done, and the data are of some interest. However, there are some major questions, which need to be addressed by the authors, also by additional data/experiments.
11) It is not clear what the overall aim of this study is. Do the authors aim to have novel compounds which inhibit both vitamin K-effects and platelets? Do the newly synthesized coumarins still have vitamin K-antagonist properties?
Reply: Coumarin drugs are synthesized from coumarin, and one of the important derivatives is warfarin. The anticoagulant effect of Coumarin is based on the activity of its derivatives (warfarin or dicumarol). It is estimated that the anticoagulant effect of coumarin may be carried out by participating in the antagonism of vitamin K metabolism. The compound used in our study is derived from coumarin or from coumarin isomer. Whether it has the same antagonistic function of vitamin K metabolism as warfarin is still unknown. Although it is still unknown whether these compounds have anti-vitamin K metabolism effect, they have been found to have antithrombotic ability in our in vivo experiments. As shown in the figure below, we found that compound 2 and compound 4 can prevent PHZ-induced hemolytic anemia in zebrafish embryos, suggesting that they have preventive or thrombosis. Relevant descriptions were presented in the Discussion section (line 178-187), but some data are not presented in detail because the experiment has not yet been completed.
22) Concentrations of compounds used. While the authors report the concentrations of ADP and fibrinogen in normal biochemical terms (i.e. µM), concentration of the coumarin derivatives is reported in ppm. For direct comparison, the concentration of coumarins should be given also in µM.
Reply: Thank you for your suggestion. The conversion of each compound unit (from ppm to nM) was written in line 102-104 and line 320-322. For the consistency of the test and the production of the chart, each compound in the chart is still expressed in ppm. Even so, the concentrations of each compound remain similar to each other.
33) Medical/pharmacological relevance. Multiple chemical compounds unspecifically inhibit platelet activation by ADP and other agonists, especially at concentrations higher than 5-10 µM. The first question is whether pharmacologically used coumarins reach plasma concentrations in the rage of the coumarin concentration used here ? Then, dose response curve would be helpful, ideally in comparison with established ADP receptor antagonists.
Reply: Many thanks to the reviewers for their professional advice. According to the suggestion, we explained in order: 1. The concentration of coumarin derivatives used in this experiment is 10 ppm and the concentration range is from 78 nM to 109.5 nM, and it is not as high as 5-10 μM. 2. The coumarin derivatives used in this study were at relatively low concentrations compared to the concentrations of warfarin (coumarin derivative) in patient plasma (as stated in the Discussion section, line 218-220). 3. The description of the inhibitory doses of coumarin derivatives compared to the currently commonly used P2Y1 and P2Y12 is stated in the Discussion section (line 206-212).
44) Specificity : Do the coumarin derivatives affect only platelet ADP receptors? The authors should also study other platelet agonists (thromboxane A2, thrombin, collagen etc.)
Reply: Many thanks for the reviewer's suggestion. As suggested by the reviewers, the agonist of platelet aggregation is not only ADP, but either collagen or thromboxane A2 would eventually induce the release of ADP and its downstream signaling effect to induce platelet aggregation. We therefore chose ADP as the single agonist. In order to convince the reader more, we make a text description in the Introduction section (line 73-88). In addition, we also explain the possible results in the Discussion section (line 212-217).
55) P2Y12 receptor responses include much more than down-regulation of cAMP, i.e. PI3K responses.
Reply: Thanks for the reviewer's suggestion. According to literature reports, platelet PI3K/AKT is involved in platelet apoptosis or the release of platelet internal granules. In our analysis, however, the number of platelets was not found to be significantly altered, implying that the AKT pathway is not affected by coumarin derivatives. Relevant textual descriptions were explained in the Discussion section (line 229-237).

Reviewer 3 Report
In the present study, Lu and colleagues assess the effects of different coumarin derivatives on ADP-induced platelet aggregation. This research might be of interest to the community since so far coumarin has been mostly investigated with regard to its effects on the plasmatic coagulation. However, unfortunately, the manuscript is poorly written, with regard to not only language, but more importantly regarding adequate terminology (see below for some examples). Moreover, it is not clear to me why the authors focused on ADP as only agonist and ignored other important aspects of platelet physiology.
Major points:
1) The introduction lacks an explanation for the authors’ focus on ADP as only agonist tested. Why was only ADP chosen, what about other relevant platelet agonists such as thrombin or collagen? Would coumarin derivatives act in a pathway-specific manner?
2) Unfortunately, I had some problems in understanding the authors’ text. In general, the English could be improved. More important, would, however, be to use the correct terminology. Here are some examples:
· Line 27: GPIIb/IIIa does not need to be “formed” it is already on the platelet surface. What differs between resting and activated platelets is the conformation of the receptor.
· Line 50, 51: Existing and new thrombosis are uncommon terms, I suggest to revise this.
· Line 60: platelet stage-platelet attachment? Revise
· Line 60: attachment, aggregation and coagulation
· Line 72: Platelets usually adhere to the damaged endothelium or on the exposed extracellular matrix.
· Line 73: In most of the cases receptors (upon ligand binding) result in platelet activation (the authors wrote “activate receptors”). Moreover, the chemical messengers are not called “activation”. Instead, most people would say that platelet activation is characterized by platelet degranulation and a functional up-regulation of platelet integrin receptors.
· Line 74: reads very bad and its incomplete.
· Line 75: it is not platelet emboli. Better would be to say platelet aggregates.
· Line 78: Usually the term “white or red clot” discriminates thrombi based on the amount of red blood cells that are entrapped in it.
· Line 91-92: Coumarin derivatives blocked ADP-dependent P2Y1 and P2Y12 platelet activation.
· Line 93/94: I would rephrase the sentence to: “These results indicate that coumarin derivatives inhibit platelet activation in addition to their inhibitory effect on the formation of coagulation factors.” As the latter statement is already proven and not related to the present study.
· Line 116: Please remove this sentence, albeit it is not fully incorrect (as platelets without GPIIb/IIIa would not aggregate), but in the generalized manner I would not agree.
· Line 121-123: ADP induces GPIIIa expression and GPIIb/IIIa activation? This statement is confusing and should be explained better. Also, in the text is written that “PAC-1 formation was measured”. I assume the authors meant binding of PAC-1 to activated GPIIb/IIIa?
· Line 154: activation of calcium Ions is an incorrect statement.
· Line 163-164: I think this sentence is wrong and need to be revised.
· Line 165: Signal transduction instead of information transduction
· Line 185-187: Why should aspirin be an anti-coagulation drug and not an anti-platelet drug?
3) Ppm is a very uncommon notation that is not part of the International System of Units (SI) system as its meaning is ambiguous. I recommend to change this to µM or comparable units.
4) Figure 2: Are the inhibitory effects ADP specific, or would a similar inhibition also be seen in response to collagen or thrombin? Did the authors check whether their compounds “kill the platelets” (instead of only inhibiting them)? Why did the authors incubate PRP with coumarin derivatives for 1 h? Usually inhibitors are added for 5-15 min in that kind of assays. How does a vehicle control look like? What about shear conditions or spreading of platelets on fibrinogen?
5) Figure 3: I am struggling with this figure for several reasons:
· there is no indication of the axis and the title of Fig. A is not really correct (should be anti-CD61)
· compound only traces are not display (but have to),
· 1 h with compound followed by 30 min with ADP are quite lengthy,
· What is the second peak in the curves? I would understand one in the case of PAC-1 (higher population of platelets with an activated GPIIb/IIIa), but not for anti-CD61 signals. Usually, platelet activation results in a subtle increase of GP surface abundances (due to degranulation and the mobilization of receptors from within the cell to the surface). However, in the control the platelet population appears to be negative (which cannot be true for GPIIIa). Moreover, usually the entire population should shift.
· The differences between ADP alone and ADP plus any of the compounds are minimal and cannot explain the inhibition observed in the aggregation studies. Consequently, also the statement in line 125-127 has to be revised.
6) Figure 4: Did the offers check whether the presence of the compounds (independently of ADP) affect the production of TxB2 or the release of AA?
7) Figure 5A: The calcium peaks appear very low and it is not clear what is meant with no compound? The authors have to display a “resting” and an ADP-control only. Likewise, an ADP-only control needs to be added to Fig. 5B.
8) In the discussion it is stated that: “These results indicate that many clinical side effects of coumarin derivatives may include inhibition of clotting factor formation and the activity of platelets, which lead to the deficiency of normal coagulation function.” However, this is only a speculation and not demonstrated by these results. The authors should more extensively address this point in the discussion and connect this to clinical data.
Minor points:
1) I would recommend to revise the introduction (line 38-42) and focus more on coumarin with regard to the present study. Of course, it is of relevant as spice and in cosmietics industry, however, cosmetics is mentioned twice. Then, it is described both as cancerogenic and as cancer-inhibitor. This is quite confusing.
2) The introduction of the plasmatic coagulation (line 57-69) is hard to read and should be revised. Moreover, usually thromboplastin is one of the first steps to trigger the coagulation cascade.
3) Figure 2A: The numbers relative to the compounds in the curve cannot be properly read, also the legend is confusing – please revise. Furthermore, I suggest to use same color coding for Fig. 2A and 2B.
4) It should be mentioned somewhere that TxB2 is the stable analogue of TxA2.
5) Line 269: 1500 g to spin down platelets seem too much and might risk platelet pre-activation
6) Line 278: Why 1 hour of incubation?
7) Line 284: What is the code/clone of the anti-CD61 antibody?
8) Line 290-294: Why the compounds were added to PRP and not to washed platelets?
Author Response
Response to Reviewer 3:
In the present study, Lu and colleagues assess the effects of different coumarin derivatives on ADP-induced platelet aggregation. This research might be of interest to the community since so far coumarin has been mostly investigated with regard to its effects on the plasmatic coagulation. However, unfortunately, the manuscript is poorly written, with regard to not only language, but more importantly regarding adequate terminology (see below for some examples). Moreover, it is not clear to me why the authors focused on ADP as only agonist and ignored other important aspects of platelet physiology.
Major points:
1) The introduction lacks an explanation for the authors’ focus on ADP as only agonist tested. Why was only ADP chosen, what about other relevant platelet agonists such as thrombin or collagen? Would coumarin derivatives act in a pathway-specific manner?
Reply: Many thanks for the reviewer's suggestion. As suggested by the reviewers, the agonist of platelet aggregation is not only ADP, but either collagen or thromboxane A2 would eventually induce the release of ADP and its downstream signaling effect to induce platelet aggregation. We therefore chose ADP as the single agonist. In order to convince the reader more, we make a text description in the Introduction section (line 73-88). In addition, we also explain the possible results in the Discussion section (line 212-217). Since coumarin derivatives all inhibit the downstream signaling of ADP, suggesting that they are less specific than the antagonist of P2Y1 or P2Y12, relevant description were presented in the Discussion section (206-212).
2) Unfortunately, I had some problems in understanding the authors’ text. In general, the English could be improved. More important, would, however, be to use the correct terminology. Here are some examples:
Reply: Thanks for the editor's suggestion. Our revised manuscript has been edited by a professional English editing company.
- Line 27: GPIIb/IIIa does not need to be “formed” it is already on the platelet surface. What differs between resting and activated platelets is the conformation of the receptor.
Reply: The text description has been revised (line 27-28).
- Line 50, 51: Existing and new thrombosis are uncommon terms, I suggest to revise this.
Reply: Thanks for the reviewer's suggestion. This content has been removed due to substantial revision of the manuscript content.
- Line 60: platelet stage-platelet attachment? Revise
Reply: The text description has been revised (line 52-53).
- Line 60: attachment, aggregation and coagulation
Reply: The text description has been revised (line 53-55).
- Line 72: Platelets usually adhere to the damaged endothelium or on the exposed extracellular matrix.
Reply: The text description has been revised (line 65-66).
- Line 73: In most of the cases receptors (upon ligand binding) result in platelet activation (the authors wrote “activate receptors”). Moreover, the chemical messengers are not called “activation”. Instead, most people would say that platelet activation is characterized by platelet degranulation and a functional up-regulation of platelet integrin receptors.
Reply: The text description has been revised (line 66-68).
- Line 74: reads very bad and its incomplete.
Reply: The text description has been revised (line 68-70).
- Line 75: it is not platelet emboli. Better would be to say platelet aggregates.
Reply: The text description has been revised according to the reviewer’s comment (line 68).
- Line 78: Usually the term “white or red clot” discriminates thrombi based on the amount of red blood cells that are entrapped in it.
Reply: The text description has been revised (line 71-73).
- Line 91-92: Coumarin derivatives blocked ADP-dependent P2Y1 and P2Y12 platelet activation.
Reply: The text description has been revised (line 94-95).
- Line 93/94: I would rephrase the sentence to: “These results indicate that coumarin derivatives inhibit platelet activation in addition to their inhibitory effect on the formation of coagulation factors.” As the latter statement is already proven and not related to the present study.
Reply: The text description has been revised according to the reviewer’s comment (line 95-96).
- Line 116: Please remove this sentence, albeit it is not fully incorrect (as platelets without GPIIb/IIIa would not aggregate), but in the generalized manner I would not agree.
Reply: The sentence was removed according to the reviewer’s comment.
- Line 121-123: ADP induces GPIIIa expression and GPIIb/IIIa activation? This statement is confusing and should be explained better. Also, in the text is written that “PAC-1 formation was measured”. I assume the authors meant binding of PAC-1 to activated GPIIb/IIIa?
Reply: Thanks for the reviewer's suggestion. The original narrative was erroneous and confusing for the reader, so we have revised this paragraph (line 122-130).
- Line 154: activation of calcium Ions is an incorrect statement.
Reply: The text description has been revised (line 157).
- Line 163-164: I think this sentence is wrong and need to be revised.
Reply: The text description has been revised (line 165-168).
- Line 165: Signal transduction instead of information transduction
Reply: The text description has been revised according to the reviewer’s comment (line 168).
- Line 185-187: Why should aspirin be an anti-coagulation drug and not an anti-platelet drug? Reply: We are very sorry that the original description was confusing. This paragraph has been revised (line 196-198).
3) Ppm is a very uncommon notation that is not part of the International System of Units (SI) system as its meaning is ambiguous. I recommend to change this to µM or comparable units. Reply: Thank you for your suggestion. The conversion of each compound unit (from ppm to nM) was written in line 104-105 and line 321-323. For the consistency of the test and the production of the chart, each compound in the chart is still expressed in ppm. Even so, the concentrations of each compound remain similar to each other.
4) Figure 2: Are the inhibitory effects ADP specific, or would a similar inhibition also be seen in response to collagen or thrombin? Did the authors check whether their compounds “kill the platelets” (instead of only inhibiting them)? Why did the authors incubate PRP with coumarin derivatives for 1 h? Usually inhibitors are added for 5-15 min in that kind of assays. How does a vehicle control look like? What about shear conditions or spreading of platelets on fibrinogen?
Reply: Thanks for the reviewer's suggestion. 1. Whether other agonists other than ADP have similar effects will be discussed in the Discussion (line 212-217). 2. By analyzing the population of platelets and the expression of CD61 on the membrane by flow cytometry, we assume that coumarin derivatives do not cause apoptosis of platelets. The relevant description is introduced in the line 232-235. 3. As professionally suggested by the reviewers, in general, aggregation assays are usually completed within 1 h after PRP isolation. Because the platelet aggregation analyzer contains eight channels and has three groups of culture microenvironments, in each of experiments, in order to reduce the time difference between each group and the feasibility of the experiment, the effect of coumarin derivatives in the PRP effect is usually between 20-40 minutes, plus stimulation with ADP for 6 minutes. The overall time is within a reasonable range, and each set of experiments was reproducible. The original time description was calculated in terms of overall time and has now been changed to 30 minutes to avoid confusion. Finally, the platelet aggregation analysis was performed on a platelet analyzer regardless of culture or drug addiction. The control group used DMSO, a diluent of coumarin derivatives, and fibrinogen was not added in the experiment, and the morphology of platelets could not be known.
5) Figure 3: I am struggling with this figure for several reasons:
- there is no indication of the axis and the title of Fig. A is not really correct (should be anti-CD61)
- compound only traces are not display (but have to),
- 1 h with compound followed by 30 min with ADP are quite lengthy,
- What is the second peak in the curves? I would understand one in the case of PAC-1 (higher population of platelets with an activated GPIIb/IIIa), but not for anti-CD61 signals. Usually, platelet activation results in a subtle increase of GP surface abundances (due to degranulation and the mobilization of receptors from within the cell to the surface). However, in the control the platelet population appears to be negative (which cannot be true for GPIIIa). Moreover, usually the entire population should shift.
- The differences between ADP alone and ADP plus any of the compounds are minimal and cannot explain the inhibition observed in the aggregation studies. Consequently, also the statement in line 125-127 has to be revised.
Reply: Thank you reviewer for your professional advice. The method used in the 1st manuscript was to perform preliminary screening based on cell size populations and directly analyze anti-CD61 and PAC-1. We re-performed this experiment with minor modifications (30 min incubation), selecting on the basis of cell size, and the distribution of CD61 expression still had two populations (fewer populations on the left). Based on the distribution of isotype antibody staining, we selected the expression groups of CD61 for subsequent analysis, as shown in the figure below. The experimental results showed that the addition of ADP still induced the expression of CD61 on the platelet membrane, while the coumarin derivatives inhibited the activated state of GPIIb/IIIa on the platelet membrane. Figure 3 was the revised version, with the relevant text in the Results section (line 119-130).
6) Figure 4: Did the offers check whether the presence of the compounds (independently of ADP) affect the production of TxB2 or the release of AA?
Reply: In this experiment we have previously tested whether coumarin derivatives affect platelet AA and TXB2 production. However, in the absence of ADP, platelets produced very little AA and TXB2, and the differences caused by coumarin derivatives could not be observed in the experiments. Related description was introduced in the Result section (line 139-141).
7) Figure 5A: The calcium peaks appear very low and it is not clear what is meant with no compound? The authors have to display a “resting” and an ADP-control only. Likewise, an ADP-only control needs to be added to Fig. 5B.
Reply: Sorry for the inconvenience to the reviewers due to the lack of graphic text descriptions. The vertical axis in Figure 5A is the fold change compared to the control group without ADP. We have described in the Figure 5A legend (line 171-173). Although our results are relatively low compared to previous studies (Zhao T., et al. 2017. Front. Pharmacol. 8:361.) where ADP can be effectively increased by about 3-fold, the possible reason is that the analytical methods and reagents are different. In Figure 5B, we have previously conducted a preliminary experiment with only ADP and no forskolin. Compared with the control group not treated with any drug, there was no significant difference in the production of platelet cAMP caused by ADP (the baseline value was too low, and the relative error was large). These results imply that platelets need to induce the production of cAMP with forskolin before comparison between different groups can be made.
8) In the discussion it is stated that: “These results indicate that many clinical side effects of coumarin derivatives may include inhibition of clotting factor formation and the activity of platelets, which lead to the deficiency of normal coagulation function.” However, this is only a speculation and not demonstrated by these results. The authors should more extensively address this point in the discussion and connect this to clinical data.
Reply: The text description has been revised according to the reviewer’s comment (line 200-204).
Minor points:
1) I would recommend to revise the introduction (line 38-42) and focus more on coumarin with regard to the present study. Of course, it is of relevant as spice and in cosmietics industry, however, cosmetics is mentioned twice. Then, it is described both as cancerogenic and as cancer-inhibitor. This is quite confusing.
Reply: Thanks for your advice. This paragraph has been removed.
2) The introduction of the plasmatic coagulation (line 57-69) is hard to read and should be revised. Moreover, usually thromboplastin is one of the first steps to trigger the coagulation cascade.
Reply: Thanks for your advice. This paragraph has been revised (line 54-63).
3) Figure 2A: The numbers relative to the compounds in the curve cannot be properly read, also the legend is confusing – please revise. Furthermore, I suggest to use same color coding for Fig. 2A and 2B.
Reply: I very much agree with the reviewer's suggestion. However, the data (Figure 2A) from light transmittance aggregometer (PAP-8E) were outputted and paste as a figure file (JPG file). The lines in the aggregometer are built-in, and there no any way to edit theses detail (such as the width and the color of line) in these figures according to the aggregometer’s instruction. Also, the figure legend has been revised (line 115-118)
4) It should be mentioned somewhere that TxB2 is the stable analogue of TxA2.
Reply: The text description has been added according to the reviewer’s comment (line 136).
5) Line 269: 1500 g to spin down platelets seem too much and might risk platelet pre-activation
Reply: Thanks for the reviewer's suggestion. Centrifugal conditions we previously referred to previous studies ( Li, JZ., et al. 1983. Thrombosis and Haemostasis. 59 (3) 435-439 (1983); Michael S Cran., et ak. 2005. Br J Pharmacol. 144(6): 849–859). The conditions for the separation of PPP and platelets by centrifugation of PRP are 2000 g 15 min and 1200 g 10min, respectively. Indeed, as suggested by the reviewer, too high a rotational speed may cause platelet activation. However, when we previously centrifuged at 1200 g, the platelet pellet that was not tightly attached was often resuspended when PPP was extracted, resulting in a high background value of serum transmittance. That's why we used 1500 g to separate PPP and platelets.
6) Line 278: Why 1 hour of incubation?
Reply: Thanks for the reviewer's suggestion. We have already explained in the fourth point above.
7) Line 284: What is the code/clone of the anti-CD61 antibody?
Reply: The name of the clone of the anti-CD61 has been added (line 332).
8) Line 290-294: Why the compounds were added to PRP and not to washed platelets?
Reply: Drug treatment, some literature test descriptions are added to purified platelets, and some are in PRP (eg: Jin J., et al. 2002. Blood. 99: 193-198; Hla Nu Swe., et al. Int. J. Mol. Sci. 2021, 22, 6846.). Since we previously used PRP for aggregation analysis, we used PRP as the test object to add various compounds, and then isolate purified platelets for subsequent experiments.

Round 2
Reviewer 2 Report
The authors responded to most comments made by referees 1-3 and made important changes in the paper. There are 2 topics which still require a more solid response than given now:
a) do the coumarin derivatives they study retain Vit.-K antagonist property?
b) do the compounds inhibit platelet receptors other than P2Y12/ P2Y1 ?
These answers could be given with moderate experimental efforts.
Author Response
Response to Reviewer 2
The authors responded to most comments made by referees 1-3 and made important changes in the paper. There are 2 topics which still require a more solid response than given now:
- do the coumarin derivatives they study retain Vit.-K antagonist property?
Reply: Thanks for the reviewer's suggestion. Exploring the analysis of vitamin K antagonists through the literature. It can be classified as radioactive substances to calibrate phylloquinone oxide as a metabolite of vitamin K for analysis (eg John T. Matschiner and Robert G. Bell. 1972. The Journal of Nutrition, 102: 625–629.; Robert G. Bell, and John T. Matschiner. 1972. Nature, 237:32–33.) or plasma collection for analysis by HPLC (e.g. Yanni Mi, et al. Int J Exp Pathol. 2016 Apr; 97(2): 187–193.; Eva Klapkova, et al . J Clin Lab Anal. 2018 Jun; 32(5): e22381.). Regardless of the method mentioned above, it is necessary to administer the tested drug (compound) in animals for a period of time, and then collect plasma for subsequent analysis.
At present, there is no in vitro measurement method in the literature, and with reference to the aforementioned studies, the time required for the application and testing of experimental animals cannot be completed in a short time, and the required measurement equipment and materials cannot be achieved in our current laboratory. Furthermore, the trials may deviate from the topic we are discussing. Even though our previous in vivo experiments can only partially explain the potential activity of coumarin derivatives (line 197-200), the above explanations hope that the reviewers will understand why we were unable to perform the experiment requested by the reviewers.
- b) do the compounds inhibit platelet receptors other than P2Y12/ P2Y1 ?
Reply: Thank you reviewer for your professional advice. We used P2Y1 and P2Y12 antagonists to block P2Y12/ P2Y1 receptors signaling, and then explored whether coumarin derivatives could inhibit platelet aggregation. It was found that when coumarin derivatives and P2Y1 and P2Y12 antagonists were added to platelets at the same time, compared with the addition of only P2Y1 and P2Y12 antagonists, there was no further inhibition of ADP-induced platelet aggregation, suggesting that coumarin derivatives regulates platelet activity through these two ADP receptors. The relevant result was shown in Figure S2 and described in the Result section (line 163-167).
These answers could be given with moderate experimental efforts.

Reviewer 3 Report
The authors‘ only addressed some of my comments and I still consider the quality of the study as problematic.
Major points:
1) Despite the fact that ADP is a relevant second wave mediator of platelet activation downstream of many platelet agonists, it does not exclude that the effect of coumarin derivatives display additional effects (beyond those observable in response to ADP) if other platelet agonists are used. In my opinion testing thrombin and a GPVI agonist is mandatory for this study.
2) Albeit language clearly improved, a few points still need to be addressed:
- line 35f.: change “in addition” to “by”
- Line 56: please revise
- Line 61: Revise, the sentence does not contain information, but it has citations.
- Line 92-94: It should be added that the “white clot” is platelet-rich
- Line 130: please revise
- Line 220-223: Please clarify whether you are talking about the compounds used in the current study or of other compounds.
3) In their response the authors stated that DMSO treatment was used as control. However, I could not spot this in the text. Instead, the legend of Fig. 2 claims that ADP only (so no DMSO or anything) was used as a control. In my hands, DMSO usually has a subtle, but measurable effect on platelet aggregation. Thus, I consider this control to be essential. Some data on the overall morphology of the coumarin-treated platelets would also be valuable.
4) Figure 3 is much clearer now, albeit I still do not understand how the data from the old Fig. 3 results in the new data. In addition, I am wondering whether the authors assessed P-selectin exposure or any other degranulation marker? The CD61 signal indicates that there is an increase for almost every coumarin-derivative tested. This usually is the consequence of platelet degranulation (which mobilizes more CD61 to the platelet surface). However, it is rarely the case that platelet degranulation is enhanced, while integrin activation decreased. This ‘mystery’ should be resolved by the auhors.
5) Fig. 5A would benefit from representative curves and Fig. 5B should also include compounds without forskolin and, actually, also the compounds without ADP.
Minor points:
1) n-numbers should be provided in the legend of Fig. 2 and Fig. 4.
Author Response
Response to Reviewer 3
The authors‘ only addressed some of my comments and I still consider the quality of the study as problematic.
Major points:
- Despite the fact that ADP is a relevant second wave mediator of platelet activation downstream of many platelet agonists, it does not exclude that the effect of coumarin derivatives display additional effects (beyond those observable in response to ADP) if other platelet agonists are used. In my opinion testing thrombin and a GPVI agonist is mandatory for this study.
Reply: Thanks for the reviewer's suggestion. At present, the platelet aggregation analyzer (PAP-8E) we used has built-in modes of ADP, collagen and AA as the analysis of platelet aggregation. At present, there is only collagen in our laboratory, so we can only use collagen as platelet agonists for follow-up analysis and please understand that we cannot use other agonists for analysis according to your suggestion. As shown in Figure 2 and Figure S1, whether ADP or collagen were used as platelet agonists, coumarin derivatives have similar inhibitory effects. According to the analysis of ADP receptor antagonists, coumarin derivatives inhibit platelet aggregation by inhibiting the signal transduction of P2Y1 and P2Y12 receptors. The relevant result was shown in Figure S1 and Figure S2 and described in the Result section (line 107-110 and line 163-167). Although we did not perform aggregation assays with thrombin or GPVI, the results of the assay with ADP receptor antagonists should explain the effect of coumarin derivatives mainly on ADP receptors.
2) Albeit language clearly improved, a few points still need to be addressed:
- line 35f.: change “in addition” to “by”
Reply: The text description has been revised (line 32).
- Line 56: please revise
Reply: The text description has been revised (line 46-47).
- Line 61: Revise, the sentence does not contain information, but it has citations.
Reply: Thank you for your suggestions. This sentence has no meaning in the article, so we have deleted it.
- Line 92-94: It should be added that the “white clot” is platelet-rich
Reply: The text description has been revised according to the reviewer’s comment (line 72-74).
- Line 130: please revise
Reply: The text description has been revised (line 106-107).
- Line 220-223: Please clarify whether you are talking about the compounds used in the current study or of other compounds.
Reply: The text description has been revised (line 201-204).
3) In their response the authors stated that DMSO treatment was used as control. However, I could not spot this in the text. Instead, the legend of Fig. 2 claims that ADP only (so no DMSO or anything) was used as a control. In my hands, DMSO usually has a subtle, but measurable effect on platelet aggregation. Thus, I consider this control to be essential. Some data on the overall morphology of the coumarin-treated platelets would also be valuable.
Reply: Thank you for your professional advice. Indeed, as stated by the reviewers, DMSO (6%) has been shown to affect platelet aggregation according to previous studies (Julien Guillaumin., et al., Journal of Veterinary Emergency and Critical Care 20(6) 2010, pp 571–577). However, when diluting coumarin derivatives, DMSO has to be used as the diluent reagent due to solubility. The highest DMSO dose concentration in our study was 1/400 (0.25%) (line 334), and this dose in our study did not differ significantly in the ADP-induced platelet aggregation assay compared to the control group without DMSO (as shown in the figure below. There are only text descriptions in the system, please refer to the PDF file attached to the reply, which has graphic descriptions). Additionally, the Figure 2 legend had be revised according to the reviewer's suggestion (line 119-120).
4) Figure 3 is much clearer now, albeit I still do not understand how the data from the old Fig. 3 results in the new data. In addition, I am wondering whether the authors assessed P-selectin exposure or any other degranulation marker? The CD61 signal indicates that there is an increase for almost every coumarin-derivative tested. This usually is the consequence of platelet degranulation (which mobilizes more CD61 to the platelet surface). However, it is rarely the case that platelet degranulation is enhanced, while integrin activation decreased. This ‘mystery’ should be resolved by the auhors.
Reply: Thank you for your professional advice. The result in revised Figure 3 is the result of the re-test, not the original figure. In addition, only text descriptions can be made in the system, and I am very sorry for the misunderstanding caused by the reviewers. Please refer to our PDF file again, the file contains graphics to make it easier for reviewers to understand the content.
In the original experiment, we used the cell size and granularity as the cell grouping in the flow cytometer, so the selected group can be divided into two groups in the anti-CD61 analysis. In our revised manuscript, in addition to the aforementioned methods, we used iso-type Ab to select anti-CD61+ populations, as shown in the figure below. After secondary selection, new CD61+ groups were obtained, and subsequent analysis was carried out.
In experiments with three different individuals, we did find that the addition of ADP resulted in a slight increase in anti-CD61 intensity on platelet membranes (as shown in the figure below, the increase in anti-CD61 intensity was induced by ADP but not coumarin derivatives). After exploring the literature, it is known that most of them discuss the number of CD61+ groups, and few literatures discuss the fluorescence intensity of anti-CD61. Few studies have shown that ADP increases the intensity and population of CD61+, CD41a, or CD62P (eg, Thomas A. Blair et al., Sci Rep. 2018; 8: 10300.), but the reasons for this remain unclear. Therefore, we are also unable to reply to this reviewer's question, and we are very sorry.
5) Fig. 5A would benefit from representative curves and Fig. 5B should also include compounds without forskolin and, actually, also the compounds without ADP.
Reply: We used the fluorescent dye Fura-2AM and EIA reader to read 340/380 values to estimate the changes of calcium ions in platelets. EIA only displays numerical values (line 369-374), and there is no special machine for reading the changes of calcium ions (such as flex mode of FlexStation) to display the graph of calcium ion changes. Additionally, coumarin derivatives appear to increase platelet cAMP levels as shown in Fig. 5B. However, our other experimental results found that the addition of coumarin derivatives did not change the content of cAMP in platelets without the addition of forskolin and ADP. The relevant test results are shown in the figure below (please refer to the PDF file) and described in the Result section (line 177-181).
Minor points:
1) n-numbers should be provided in the legend of Fig. 2 and Fig. 4.
Reply: The text description has been revised according to the reviewer comment (line 122 and line 156).
